# Risk of major adverse cardiovascular events with aripiprazole versus olanzapine, quetiapine, and risperidone in severe mental illness: a target trial emulation

Alvin Richards-Belle [1] ✉, Naomi Launders [1], Sarah Hardoon[1], Kenneth K. C. Man [2,3,4,5], Neil M. Davies [1,6,7], Elvira Bramon [1,8], Joseph F. Hayes[1,8] & David P. J. Osborn[1,8]

Initiating aripiprazole as antipsychotic monotherapy rather than olanzapine, quetiapine, or risperidone, might prevent/delay major adverse cardiovascular events (MACEs) over the long-term in people diagnosed with severe mental illness. Using Clinical Practice Research Datalink data, we emulated a trial of aripiprazole versus olanzapine, quetiapine, and risperidone in 20,404 patients 2005–2014. Primary outcome was five-year MACE risk (composite of hospitalisation for acute myocardial infarction or stroke and cardiovascular death). Here we show that patients initiating aripiprazole had a similar five-year MACE risk as those initiating olanzapine (risk ratio: 1.03, 95% CI, 0.78-1.32), quetiapine (1.02, 95% CI, 0.72-1.32), and risperidone (0.88, 95% CI, 0.67-1.17). Risk was lower among patients initiating and continuing aripiprazole versus risperidone (0.58, 95% CI, 0.39-0.84). For patients at clinical equipoise, antipsychotic selection does not appear to significantly impact risk of the most severe, long-term cardiovascular events. However, further research is needed to replicate our finding of increased risk with continued risperidone use versus aripiprazole.

People diagnosed with severe mental illnesses (SMI) have premature mortality, with cardio-cerebrovascular morbidity a leading cause[1]. Their risk of cardiovascular disease is 78% higher than in the general population[1]. Antipsychotics are effective for many patients but can cause adverse cardiometabolic effects, including obesity, hyperglycaemia, and dyslipidaemia[2–5]. These effects increase risks for major adverse cardiovascular events (MACEs), such as myocardial infarction and stroke[1,6]. A comprehensive meta-analysis of observational cohort studies reported that use of second-generation antipsychotics (versus non-use) is protective overall for cardio-cerebrovascular mortality[7]. Despite this class-level effect, the varying cardiometabolic profiles of second-generation antipsychotics suggest potential differences in long-term cardio-cerebrovascular risk and that optimising prescribing might mitigate burden.

Randomised clinical trials (RCTs) and observational studies of the most frequently prescribed antipsychotics in the UK - aripiprazole,

[1]Division of Psychiatry, University College London, Maple House, London, UK. [2]Research Department of Practice and Policy, School of Pharmacy, University College London, London, UK. [3]Centre for Medicines Optimisation Research and Education, University College London Hospitals NHS Foundation Trust, London, UK. [4]Department of Pharmacology and Pharmacy, Li Ka Shing Faculty of Medicine, University of Hong Kong, Hong Kong, China. [5]Laboratory of Data Discovery for Health (D24H), Hong Kong Science Park, Hong Kong, China. [6]Department of Statistical Sciences, University College London, London, UK. [7]Department of Public Health and Nursing, Norwegian University of Science and Technology, Trondheim, Norway. [8]North London NHS Foundation Trust, London, UK. ✉e-mail: alvin.richards-belle@ucl.ac.uk

olanzapine, quetiapine, and risperidone[8] - indicate broadly similar effectiveness for reducing SMI symptoms, but that aripiprazole has better short- and medium-term cardiometabolic outcomes, such as weight gain[2–5,9]. We hypothesised that initiating aripiprazole as monotherapy, rather than olanzapine, quetiapine, or risperidone, might prevent or delay MACE over the long term.

An RCT is the gold standard methodology to investigate our hypothesis, but none have studied MACE as a primary outcome, and such a trial would be challenging to conduct due to sample size, follow-up length, adherence, and cost. Observational studies using real-world data offer an alternative, more timely, source of evidence, and typically allow for larger sample sizes and longer follow-up than RCTs. One claims study reported lower cardiovascular event incidence among patients initiating and continuing aripiprazole versus olanzapine, quetiapine, and risperidone[10], but most (>80%) patients had major depressive disorder or "other" psychiatric conditions, typically requiring lower doses and shorter treatment durations than SMI. To fill this evidence gap, we aimed to emulate an RCT addressing our hypothesis using linked electronic health records. We built on earlier work demonstrating the feasibility of emulating RCTs of antipsychotics in SMI using linked UK data sources and which generated clinically important and credible real-world evidence, including replication of the established weight gain observed with olanzapine versus aripiprazole[2].

## Results
### Sample
A total of 20,404 patients met eligibility criteria and were included (aripiprazole: $n = 1807$, olanzapine: $n = 7965$, quetiapine: $n = 5613$, risperidone: $n = 5019$) (Fig. 1). Patients initiating aripiprazole were similar in age and sex to those initiating comparators, but more likely to be from an ethnic minority, diagnosed with schizophrenia, have comorbid diabetes, and have body mass index ≥30 (Supplementary Table 1). Starting equivalent daily doses were similar for patients initiating aripiprazole and olanzapine, but lower for quetiapine and risperidone.

Median follow-up time for each of the four antipsychotic groups was 5 years. By 5 years, 1116 patients had experienced MACE, 2081 died of non-cardiovascular causes, and 17,207 completed the study uncensored. Five-year MACE incidence was 5.76% (95% CI, 5.43% to 6.09%) with a median (IQR) time to first event of 27 (12–45) months.

### Overlap weighted population
Following multiple imputation and generalised overlap weighting, the mean effective sample size across imputed datasets was 11,129 (aripiprazole: $n = 1439$, olanzapine: $n = 4542$, quetiapine: $n = 2238$, risperidone: $n = 2910$). Baseline characteristics of the weighted population are shown in Table 1. Mean standardised differences across imputed datasets between patients initiating aripiprazole versus each comparator were ≤0.10 for all baseline covariates (Supplementary Fig. 1; Supplementary Table 2). The distribution of mean weights across treatments overlapped (Supplementary Fig. 2).

### Intention-to-treat analysis
The estimated 5-year MACE risk with aripiprazole initiation corresponded to 50.6 (95% CI, 37.7 to 56.6) events for every 1000 patients treated (Supplementary Table 3). Comparatively, patients initiating aripiprazole had similar risks to those initiating olanzapine (risk difference per 1000: 1.2 events, 95% CI, −12.0 to 12.7), quetiapine (0.6 events, 95% CI, −16.7 to 12.3), and risperidone (−7.0 events, 95% CI, −20.1 to 7.2) (Table 2, Fig. 2). Risks were also similar for 6-month MACE and 5-year MACE components (Table 2).

We found strong evidence of effect modification in 5-year MACE risk by age (Wald $p < 0.001$), ethnicity ($p = 0.014$), starting dose ($p = 0.003$), but not SMI diagnosis ($p = 0.051$) or sex ($p = 0.221$). In stratified analyses, patients aged <55 years at baseline initiating aripiprazole had lower risk than those initiating risperidone (risk ratio, 0.60, 95% CI, 0.32 to 0.91). Point estimates varied across other stratified analyses, but confidence intervals were wide and mostly overlapped within subgroups (Fig. 3).

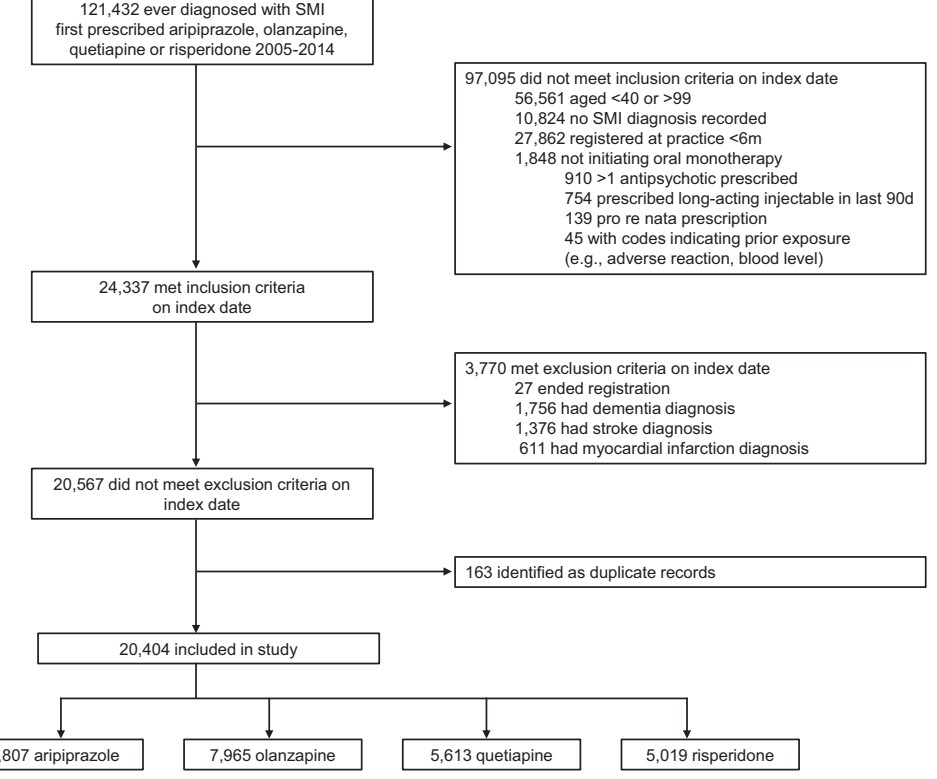

**Fig. 1 | Study flow diagram.** SMI severe mental illness.

**Table 1 | Baseline characteristics in the overlap weighted population**

| Characteristic | Aripiprazole N = 1439 | Olanzapine N = 4542 | Quetiapine N = 2238 | Risperidone N = 2910 |
|---|---|---|---|---|
| Age (years), median (IQR) | 52 (45, 63) | 52 (45, 64) | 52 (45, 62) | 52 (45, 62) |
| Sex, n (%) | | | | |
| Female | 813 (56.5%) | 2593 (57.1%) | 1263 (56.5%) | 1600 (55.0%) |
| Male | 625 (43.5%) | 1949 (42.9%) | 974 (43.5%) | 1310 (45.0%) |
| Ethnicity, n (%) | | | | |
| Asian | 94 (6.5%) | 290 (6.4%) | 144 (6.4%) | 200 (6.9%) |
| Black | 100 (6.9%) | 308 (6.8%) | 146 (6.5%) | 210 (7.2%) |
| Mixed/Other | 42 (2.9%) | 141 (3.1%) | 72 (3.2%) | 85 (2.9%) |
| White | 1204 (83.7%) | 3804 (83.7%) | 1876 (83.8%) | 2415 (83.0%) |
| SMI diagnosis, n (%) | | | | |
| Bipolar disorder | 523 (36.3%) | 1630 (35.9%) | 808 (36.1%) | 1041 (35.8%) |
| Other non-organic psychoses | 532 (37.0%) | 1734 (38.2%) | 864 (38.6%) | 1092 (37.5%) |
| Schizophrenia | 384 (26.7%) | 1178 (25.9%) | 566 (25.3%) | 777 (26.7%) |
| Age at SMI diagnosis, median (IQR) | 44 (36, 54) | 44 (35, 54) | 44 (36, 53) | 44 (35, 54) |
| SMI diagnosis to index date (years), median (IQR) | 5 (0, 16) | 5 (0, 17) | 5 (0, 16) | 5 (0, 16) |
| Index of multiple deprivation (quintile), n (%)[1] | | | | |
| 1 (Least deprived) | 187 (13.0%) | 602 (13.2%) | 283 (12.6%) | 371 (12.8%) |
| 2 | 220 (15.3%) | 716 (15.8%) | 344 (15.4%) | 456 (15.7%) |
| 3 | 272 (18.9%) | 852 (18.8%) | 412 (18.4%) | 569 (19.6%) |
| 4 | 334 (23.2%) | 1083 (23.8%) | 545 (24.4%) | 669 (23.0%) |
| 5 (Most deprived) | 426 (29.6%) | 1290 (28.4%) | 654 (29.2%) | 844 (29.0%) |
| Index year, median (IQR) | 2010 (2008, 2012) | 2010 (2007, 2012) | 2010 (2007, 2012) | 2010 (2007, 2012) |
| Comorbidities, n (%)[2] | | | | |
| Alcohol misuse | 196 (13.6%) | 611 (13.4%) | 314 (14.0%) | 416 (14.3%) |
| Angina | 47 (3.3%) | 157 (3.5%) | 73 (3.3%) | 102 (3.5%) |
| Arrhythmia | 56 (3.9%) | 195 (4.3%) | 84 (3.8%) | 110 (3.8%) |
| Diabetes | 203 (14.1%) | 633 (13.9%) | 325 (14.5%) | 422 (14.5%) |
| Dyslipidaemia | 240 (16.7%) | 751 (16.5%) | 392 (17.5%) | 470 (16.1%) |
| Hypertension | 395 (27.5%) | 1281 (28.2%) | 623 (27.8%) | 817 (28.1%) |
| Liver disease | 59 (4.1%) | 194 (4.3%) | 86 (3.8%) | 116 (4.0%) |
| Renal disease | 133 (9.3%) | 473 (10.4%) | 213 (9.5%) | 291 (10.0%) |
| Substance misuse | 136 (9.4%) | 407 (9.0%) | 212 (9.5%) | 276 (9.5%) |
| Concomitant medications, n (%)[3] | | | | |
| Prior antipsychotic use | 365 (25.4%) | 1178 (25.9%) | 588 (26.3%) | 784 (26.9%) |
| Anticoagulants | 33 (2.3%) | 99 (2.2%) | 52 (2.3%) | 73 (2.5%) |
| Antidepressants | 813 (56.5%) | 2515 (55.4%) | 1293 (57.8%) | 1609 (55.3%) |
| Antidiabetics | 139 (9.6%) | 440 (9.7%) | 233 (10.4%) | 291 (10.0%) |
| Antihypertensives | 438 (30.5%) | 1410 (31.0%) | 678 (30.3%) | 894 (30.7%) |
| Antiplatelets | 156 (10.8%) | 485 (10.7%) | 228 (10.2%) | 329 (11.3%) |
| Anxiolytics | 75 (5.2%) | 201 (4.4%) | 106 (4.7%) | 137 (4.7%) |
| Benzodiazepines | 421 (29.3%) | 1363 (30.0%) | 665 (29.7%) | 877 (30.1%) |
| Lipid regulating medications | 292 (20.3%) | 903 (19.9%) | 447 (20.0%) | 606 (20.8%) |
| Mood stabilisers | 394 (27.4%) | 1219 (26.8%) | 625 (28.0%) | 815 (28.0%) |
| Z-drugs | 319 (22.2%) | 969 (21.3%) | 502 (22.4%) | 615 (21.1%) |
| Primary care consultations in last 6m, median (IQR) | 5 (3, 10) | 6 (3, 10) | 6 (3, 10) | 6 (3, 10) |
| Smoking status, n (%) | | | | |
| Current smoker | 659 (45.8%) | 2065 (45.5%) | 1020 (45.6%) | 1322 (45.4%) |
| Ex-smoker | 206 (14.3%) | 678 (14.9%) | 323 (14.4%) | 444 (15.3%) |
| Never smoked | 574 (39.9%) | 1800 (39.6%) | 895 (40.0%) | 1143 (39.3%) |
| BMI category, n (%) | | | | |
| Healthy | 462 (32.1%) | 1439 (31.7%) | 697 (31.2%) | 908 (31.2%) |
| Obese | 451 (31.3%) | 1468 (32.3%) | 727 (32.5%) | 962 (33.1%) |
| Overweight | 483 (33.6%) | 1513 (33.3%) | 757 (33.8%) | 959 (33.0%) |
| Underweight | 43 (3.0%) | 123 (2.7%) | 56 (2.5%) | 81 (2.8%) |
| BMI (kg/m$^2$), mean (SD) | 28.2 (6.5) | 27.7 (6.0) | 28.2 (6.4) | 28.2 (6.3) |

**Table 1 (continued) | Baseline characteristics in the overlap weighted population**

| Characteristic | Aripiprazole N = 1439 | Olanzapine N = 4542 | Quetiapine N = 2238 | Risperidone N = 2910 |
|---|---|---|---|---|
| Body weight (kg), median (IQR) | 76 (65, 90) | 76 (64, 88) | 77 (66, 90) | 76 (65, 90) |
| Psychiatric hospitalisation in prior 2 y, n (%) | 463 (32.2%) | 1365 (30.1%) | 730 (32.6%) | 916 (31.5%) |
| Physical health hospitalisation in prior 2 y, n (%) | 451 (31.4%) | 1417 (31.2%) | 726 (32.4%) | 915 (31.4%) |
| Starting daily dose (olanzapine equiv.) (mg), mean (SD)[4] | 6.5 (4.0) | 6.3 (4.0) | 6.7 (4.3) | 6.4 (4.0) |

*SMI* severe mental illness, *BMI* body mass index, *mg* milligrams, *kg* kilograms, *SD* standard deviation, *IQR* interquartile range.
Characteristics estimated based on values from one multiply imputed dataset.
[1] Quintile of the 2019 English Index of Multiple Deprivation - a small area measure of relative deprivation, defined according to the patient's residential postcode or, where this was missing (unweighted n = 42), the primary care practice postcode.
[2] Comorbidities determined at any point in the patient's medical history up to and including the index date.
[3] Concomitant medications defined according to prescriptions on, or in the two years prior to, the index date (prior antipsychotic use considered only prescriptions prior).
[4] Calculated according to the Defined Daily Dose[29] method and expressed as an olanzapine equivalent dose.

**Table 2 | MACE outcomes from the intention-to-treat analysis - risk differences, risk ratios, and hazard ratios**

| | Aripiprazole vs. Olanzapine | Aripiprazole vs. Quetiapine | Aripiprazole vs. Risperidone |
|---|---|---|---|
| *Primary outcome* | | | |
| MACE, 5 y | | | |
| RD per 1000 (95% CI) | 1.2 (−12.0, 12.7) | 0.6 (−16.7, 12.3) | −7.0 (−20.1, 7.2) |
| RR (95% CI) | 1.03 (0.78, 1.32) | 1.02 (0.72, 1.32) | 0.88 (0.67, 1.17) |
| HR[1] (95% CI) | 1.01 (0.77, 1.33) | 0.89 (0.67, 1.17) | 0.92 (0.68, 1.24) |
| HR[2] (95% CI) | 1.01 (0.76, 1.32) | 0.92 (0.68, 1.26) | 0.88 (0.66, 1.17) |
| *Secondary outcomes* | | | |
| MACE, 6 m | | | |
| RD per 1000 (95% CI) | −2.8 (−5.1, 0.3) | −3.5 (−6.2, 0.3) | −2.2 (−4.4, 0.9) |
| RR (95% CI) | 0.65 (0.35, 1.12) | 0.60 (0.32, 1.14) | 0.71 (0.38, 1.29) |
| Non-fatal stroke, 5 y | | | |
| RD per 1000 (95% CI) | 2.8 (−3.6, 7.5) | 0.6 (−6.5, 5.9) | −0.1 (−6.2, 5.6) |
| RR (95% CI) | 1.25 (0.74, 1.90) | 1.05 (0.61, 1.65) | 0.99 (0.61, 1.56) |
| Non-fatal myocardial infarction, 5 y | | | |
| RD per 1000 (95% CI) | −3.0 (−7.1, 1.0) | 0.1 (−4.8, 3.9) | −2.3 (−6.9, 2.0) |
| RR (95% CI) | 0.70 (0.34, 1.16) | 1.02 (0.49, 1.99) | 0.75 (0.37, 1.35) |
| Cardiovascular death, 5 y | | | |
| RD per 1000 (95% CI) | −0.6 (−10.2, 9.1) | −1.8 (−14.8, 8.2) | −6.3 (−16.1, 5.2) |
| RR (95% CI) | 0.98 (0.70, 1.37) | 0.95 (0.63, 1.36) | 0.83 (0.60, 1.21) |

Risk differences and ratios were computed from risks predicted using a weighted pooled logistic regression model. Multiple imputation was used to handle missing covariate data, with 25 imputed datasets generated. Point estimates were pooled across imputed datasets according to Rubin's rules. We derived bias-corrected and accelerated percentile-based 95% confidence intervals from 12,500 estimates obtained by bootstrapping the procedure for estimating weights and outcome models, using 500 samples (stratified by treatment) per imputed dataset.
[1] Approximate hazard ratios estimated from the weighted pooled logistic regression model.
[2] Hazard ratios estimated using a continuous-time Cox proportional hazards model with robust standard errors.
*MACE* major adverse cardiovascular event, *RD* risk difference, *RR* risk ratio, *HR* hazard ratio.

## Per-protocol analysis

Median weighted per-protocol follow-up time was 2.15 years (ranging from 1.87 to 2.68 in patients prescribed aripiprazole and quetiapine, respectively). By 5 years, 45.3% of patients initiating aripiprazole were estimated as no longer adherent with the original strategy, which compared to 44.1%, 38.3%, and 42.1% in those initiating olanzapine, quetiapine, and risperidone, respectively (Supplementary Table 4). Recorded adverse reactions within 30 days of switching, adding, or discontinuing the original antipsychotic were rare (aripiprazole: n = 5, olanzapine: n = 23, quetiapine: n = 26, risperidone: n = 17).

The estimated 5-year MACE risk under a strategy of initiation and continued aripiprazole prescription corresponded to 37.5 (95% CI, 21.7 to 41.7) events for every 1000 patients treated. Comparatively, patients continuing aripiprazole had similar risks as those continuing olanzapine (risk difference per 1000: −11.4 events, 95% CI, −22.0 to 3.7) and quetiapine (−4.6 events, 95% CI, −20.1 to 7.6), but a lower risk than patients continuing risperidone (−27.8 events, 95% CI, −36.5 to −7.6) (Table 3).

## Sensitivity analyses

We observed consistency in estimates and confidence intervals between the intention-to-treat primary and sensitivity analyses, including different approaches to modelling, handling confounding, complete case analysis, and using more inclusive eligibility criteria (resulting in a ~20% increased weighted mean effective sample size, n = 13,283) (Fig. 4, and Supplementary Table 5). The exception was the inverse probability of treatment weighting pooled logistic model results which indicated a lower 5-year MACE risk with aripiprazole initiation versus risperidone in the overall population—however, evidence for this effect was less strong when estimated with a weighted Cox model.

Consistent with primary per-protocol analyses, all sensitivity analyses, including with different censoring weight approaches, found a lower 5-year MACE risk with continuing aripiprazole versus risperidone. Using inverse probability of censoring weighting (with censoring weights estimated based on baseline covariates and a monthly time-varying count of cumulative comorbidities during follow-up), the

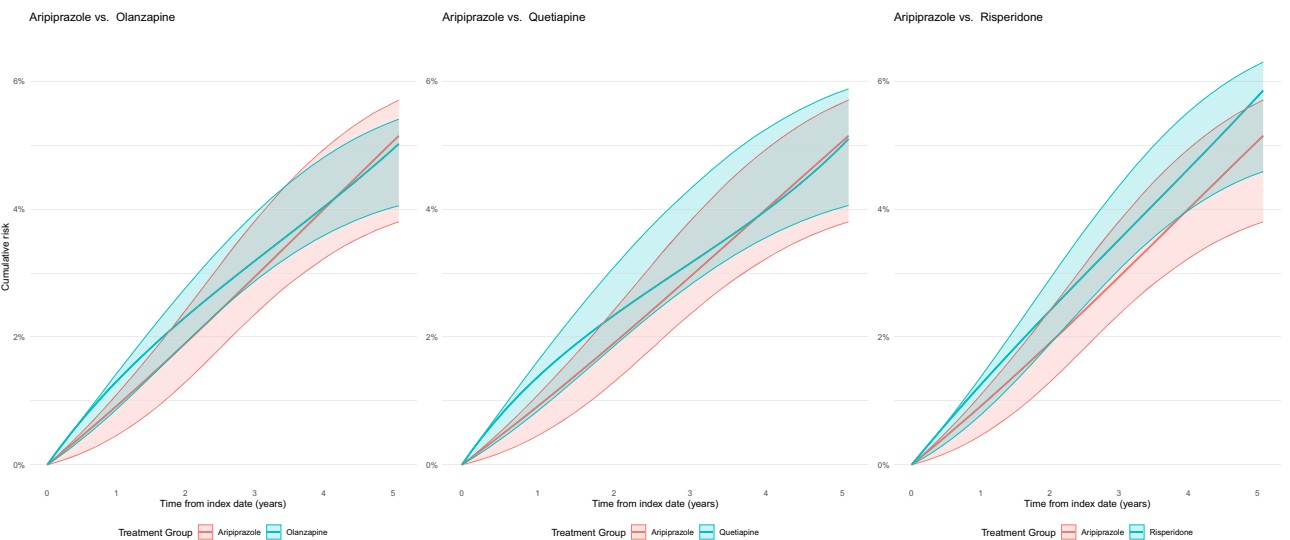

**Fig. 2 | Comparison of the cumulative risk of MACE with aripiprazole initiation versus each comparator antipsychotic from the intention-to-treat analysis.** The figure shows absolute risks with 95% confidence intervals for the aripiprazole initiation strategy versus each comparator antipsychotic. Using raw data on a total of 20,404 patients diagnosed with severe mental illness, an overlap-weighted pooled logistic regression model was used to estimate the average monthly probability of each outcome under each strategy. The model was used to predict the absolute risks. Point estimates were pooled across the 25 imputed datasets using Rubin's rules. Bias-corrected and accelerated percentile-based 95% confidence intervals were derived from 12,500 estimates obtained by bootstrapping the procedure for estimating weights and outcome models, using 500 samples (stratified by treatment) per imputed dataset. Source data are provided as a Source Data file.

5-year risk difference for patients continuing aripiprazole versus risperidone was −26.0 events per 1000 (95% CI, −35.0 to −6.5), while risks remained similar when compared to continued olanzapine (−9.6 events, 95% CI, −20.2 to 4.1) and quetiapine (−2.1 events, 95% CI, −17.3 to 9.3) use. The inverse probability of treatment weighting pooled logistic model results also indicated a lower risk with continuing aripiprazole versus quetiapine in the overall population, but with less strong evidence when estimated with a weighted Cox model (Supplementary Fig. 3, and Supplementary Table 5).

## Discussion

In this target trial emulation involving 20,404 patients, with analyses weighted toward a target population at clinical equipoise for selection of second-generation antipsychotic, we found that patients initiating aripiprazole in real-world conditions had similar 5-year risks of MACE as those initiating olanzapine, quetiapine, and risperidone. Our per-protocol analysis showed that patients continuing aripiprazole had a lower risk than those continuing risperidone. These results have several important clinical and research implications.

First, despite strong clinical belief and favourable short- to medium-term evidence on outcomes such as weight gain[2–5,9], aripiprazole may not be superior to the comparators in the longer-term risk of the more severe outcome of MACE. This could be reassuring to patients and clinicians taking or prescribing the comparators and supports ongoing flexibility and individualised clinical decision-making. That said, none of our results suggest worse outcomes with aripiprazole—in line with the few other studies investigating MACE in this area[10] and consistent with broader short- to medium-term evidence on cardiometabolic outcomes[2–5,9]. Nonetheless, evidence on intermediate cardiometabolic outcomes should still feed into decision-making, given their negative impact on patients' quality of life and medication adherence.

Second, for many patients, antipsychotic choice might play a relatively minor role in the elevated cardio-cerebrovascular risk observed in SMI. Evaluation of interventions targeting other modifiable factors, such as physical activity, diet, substances, suboptimal physical healthcare, and social determinants, are needed. These will need robust implementation efforts and will be challenging due to risk accumulation and interaction (including with medications). Exploring the association between the severity of mental illness and cardio-cerebrovascular risk is also an important area for future research.

Third, our per-protocol analysis found that patients continuing aripiprazole had a lower 5-year MACE risk than those continuing risperidone. It is not unexpected that results from intention-to-treat and per-protocol approaches differ—as they address related but distinct questions. Our intention-to-treat analyses estimated the effect of initiation and did not account for post-initiation factors—reflecting real-world levels of adherence and follow-up. In contrast, our per-protocol analyses censored patients at switch/discontinuation, which reduced the median follow-up time (from 5 to just over 2 years) and number of MACE events (from 1116 to 616). This also increased selection bias risk, but our results remained robust to potential differential censoring patterns.

Given the severity and burden that MACEs place on patients and health services—if our per-protocol finding is true, then aripiprazole might represent a more prudent choice over risperidone. To inform whether more cautious long-term risperidone prescribing is warranted, further research is required to replicate this finding and elucidate underlying mechanisms (e.g., QTc prolongation). Future research should also examine the effects of dose variation and treatment duration, as well as delineate the temporality of risk post-discontinuation, particularly since olanzapine and quetiapine are typically thought to confer greater cardiovascular risk than risperidone. Further exploration of treatment response heterogeneity could also be informative.

More generally, much of the research into the potential effects of antipsychotics on MACE has either focused on older adults (with and without dementia)[11] or the general population[12] and has investigated antipsychotic use versus non-use or use of first- versus second-generation antipsychotics[11,12]. These studies are less applicable in SMI, where a typical clinical dilemma is which second-generation antipsychotic to prescribe. In an approach most comparable to our per-protocol analysis, one of the few previous studies evaluating risks of individual antipsychotics reported that, over -1 year of follow-up,

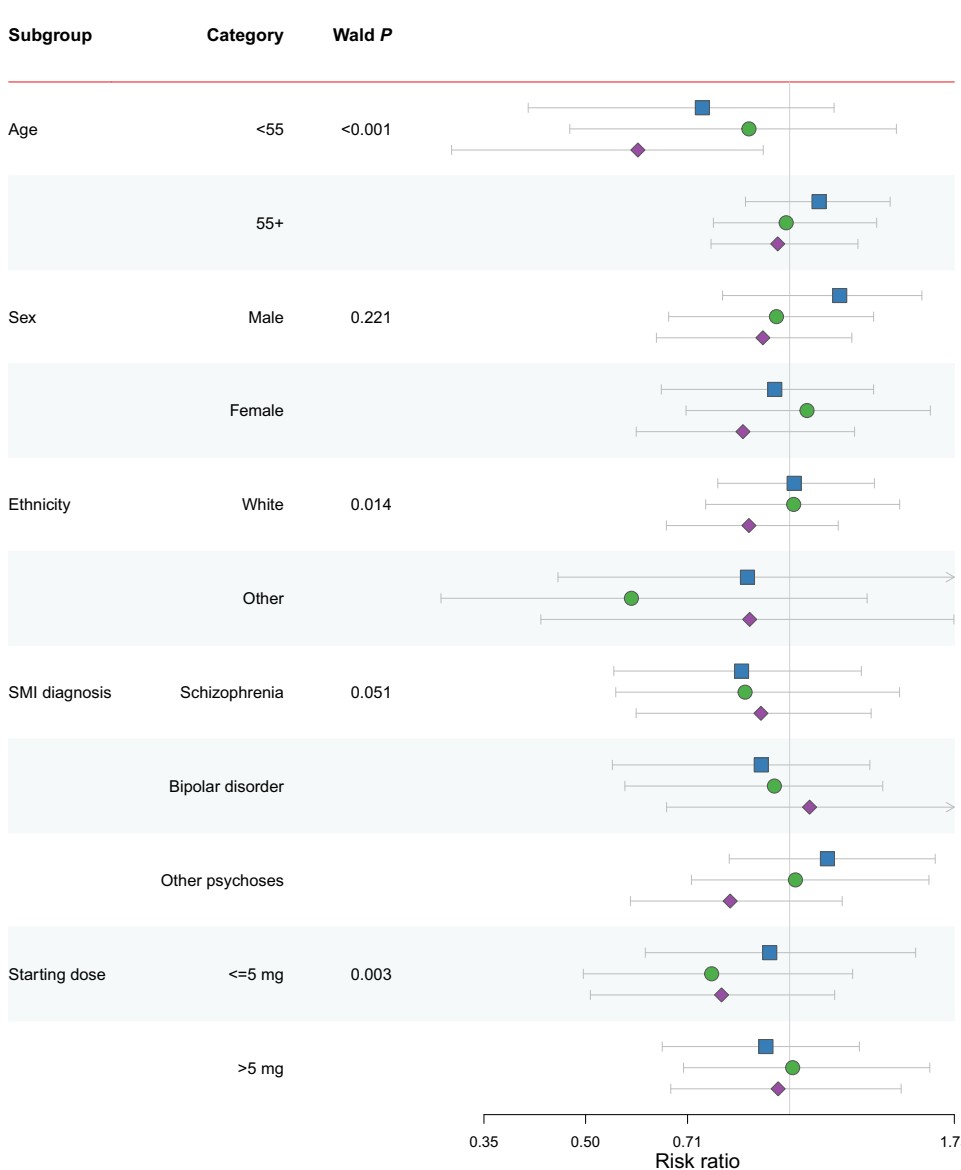

**Fig. 3 | Five-year risk of MACE with aripiprazole initiation versus each comparator antipsychotic from the intention-to-treat analysis - stratified by subgroups.** SMI severe mental illness, mg milligrams. The figure shows risk ratios with 95% confidence intervals. The unweighted total number of patients in each subgroup were as follows – age <55: $n = 10,588$; age 55+: $n = 9816$; male: $n = 8758$; female: $n = 11,646$; White: $n = 17,256$; other ethnicity: $n = 3149$; schizophrenia: $n = 4027$; bipolar disorder: $n = 8421$; other psychoses: $n = 7956$; dose <=5 mg: $n = 9433$; dose >5 mg: $n = 6804$. Two-sided Wald test $P$-values were calculated for the comparison of confounder-adjusted Cox models with and without relevant interaction terms. The exact $P$ value for age was $1.66 \times 10^{-120}$. Starting doses were calculated according to the Defined Daily Dose[29] method and expressed as an olanzapine equivalent dose. Age and starting dose were treated as continuous variables in the underlying models. In stratified analyses, age and starting dose were stratified according to median values. Source data are provided as a Source Data file.

compared to patients prescribed aripiprazole, those prescribed olanzapine (HR, 1.28, 95% CI, 1.05 to 1.55), quetiapine (1.50, 95% CI, 1.25 to 1.79), and risperidone (1.49, 95% CI, 1.21 to 1.83) had higher cardiovascular event incidence[10]. When not censoring at switch/discontinuation (comparable to our intention-to-treat analysis), results were described as consistent. Although this study was large ($n = 138,523$), few patients were diagnosed with schizophrenia (<3%) or bipolar disorder (<16%) and 43–75% of patients across comparisons were excluded through propensity score matching. Other methodological differences (e.g., follow-up duration, outcome definitions) limit comparability, but interestingly, our per-protocol effect sizes for aripiprazole versus risperidone were similar.

Our study has several strengths. First, we used real-world data on a large sample of patients from an underserved population linked from multiple national sources to evaluate longer-term risks of individual antipsychotics. To reduce ascertainment and selection bias, we focused on severe outcomes (resulting in hospital admission or death) for which there is objective national recording. This reduced the risk of selectively detecting outcomes in patients more frequently engaging in healthcare and avoided potential misclassification of prevalent events as incident. Second, building on prior work demonstrating feasibility[2], we leveraged the target trial emulation framework—minimising methodological biases and ensuring correct alignment of time zero[13]—to emulate a multi-arm trial unlikely to ever be undertaken at

**Table 3 | Five-year MACE outcome from the per-protocol analysis - risk differences, risk ratios, and hazard ratios**

|  | Aripiprazole vs. Olanzapine | Aripiprazole vs. Quetiapine | Aripiprazole vs. Risperidone |
|---|---|---|---|
| MACE, 5 y |  |  |  |
| RD per 1000 (95% CI) | −11.4 (−22.0, 3.7) | −4.6 (−20.1, 7.6) | −27.8 (−36.5, −7.6) |
| RR (95% CI) | 0.77 (0.51, 1.10) | 0.90 (0.53, 1.24) | 0.58 (0.39, 0.84) |
| HR[1] (95% CI) | 0.73 (0.49, 1.11) | 0.78 (0.50, 1.23) | 0.60 (0.39, 0.91) |
| HR[2] (95% CI) | 0.73 (0.48, 1.12) | 0.79 (0.49, 1.25) | 0.59 (0.39, 0.91) |

*MACE* major adverse cardiovascular event, *RD* risk difference, *RR* risk ratio; *HR* hazard ratio.
Risk differences and ratios were computed from risks predicted using a weighted pooled logistic regression model. Multiple imputation was used to handle missing covariate data, with 25 imputed datasets generated. Point estimates were pooled across imputed datasets according to Rubin's rules. We derived bias-corrected and accelerated percentile-based 95% confidence intervals from 12,500 estimates obtained by bootstrapping the procedure for estimating weights and outcome models, using 500 samples (stratified by treatment) per imputed dataset.
[1] Approximate hazard ratios estimated from the pooled logistic regression model.
[2] Hazard ratios estimated using a continuous-time Cox proportional hazards model with robust standard errors.

the same scale. Third, we used generalised overlap weighting—a statistically efficient approach to reducing confounding and which compares favourably to other approaches[14]—to focus on patients at clinical equipoise[14], and so inference is based on a population most likely to be treated with either of the four antipsychotics. Finally, we followed a recommended MACE definition[15].

This study also has limitations. First, as with all observational studies, results might be subject to unobserved and residual confounding—but we accounted for a wide range of confounders, including starting dose, and used a directed acyclic graph to show our assumptions. Second, we did not have data on medication dispensation or adherence, and so relied on prescription records to define treatment strategies. Third, our focus on severe outcomes means we did not include MACE events solely managed in the community. Fourth, the low event rate precluded per-protocol analysis of secondary outcomes. Fifth, our focus on patients at clinical equipoise precludes inference on patients not at clinical equipoise for whom risks might differ. Sixth, results may not generalise to settings where health service provision varies or there are differences in cardiovascular risk screening for people diagnosed with SMI. Finally, we focused on antipsychotic safety—effectiveness is the other crucial domain, within which the existing body of evidence suggests that the antipsychotics we studied are similar[2,4,5,9], making safety data vital for decision-making.

For patients at clinical equipoise for selection of a second-generation antipsychotic, those initiating aripiprazole in real-world conditions had similar 5-year risks of MACE as those initiating olanzapine, quetiapine, and risperidone. Therefore, for many patients, antipsychotic selection does not appear to have significant effects on risk of the most severe, long-term cardiovascular events. We need further research to replicate our finding of reduced risk in patients continuing aripiprazole versus risperidone and to elucidate underlying mechanisms.

## Methods
### Study design
We conducted a pre-registered[16], observational, electronic health record, target trial emulation[13] study, reported to Strengthening the Reporting of Observational Studies in Epidemiology (STROBE) guidelines[17].

Target trial emulation is a causal inference framework[13] for designing and conducting observational studies which emulate the design features of hypothetical pragmatic RCTs addressing the same causal questions. It aims to minimise biases, improve assumption transparency, and is particularly valuable in settings where rigorous RCTs are not feasible. First, we designed the protocol for the hypothetical target trial that would investigate our hypothesis. This specified that, following a clinical decision to initiate one of aripiprazole, olanzapine, quetiapine, or risperidone as monotherapy, eligible patients would be randomised to initiate one of the four antipsychotics and followed over 5 years for MACE outcomes. We then mapped the features of the target trial protocol to the design of the present

emulation (Supplementary Table 6). Two lived experience advisors were recruited to the study advisory committee and brought invaluable expertise on the real-world effects of antipsychotic medications. They were involved in study design, oversight, and interpretation.

### Data source
We used anonymised data from the Clinical Practice Research Datalink (CPRD) GOLD and Aurum databases, linked to Hospital Episode Statistics (HES) Admitted Patient Care and Office for National Statistics (ONS) death registrations. CPRD databases contain coded data (including prescriptions, diagnoses, and test results) from contributing primary care practices across the UK and are broadly representative in terms of age, sex, and ethnicity[18,19]. In England, HES captures all NHS hospital admissions[20] and ONS records all registered deaths. We used the April 2023 build of GOLD, the May 2022 build of Aurum, HES data finalised to March 2020, and death registrations up to March 2021.

CPRD has ethical approval from the East Midlands–Derby Research Ethics Committee (21/EM/0265). We obtained ethical approval for this study from CPRD's Independent Scientific Advisory Committee (protocol: 21_000729). Patient consent was not required as data are anonymised.

### Population
Patients aged 40–99, diagnosed with SMI (i.e., schizophrenia, bipolar disorder, and other non-organic psychoses)[21], without a history of stroke, myocardial infarction, or dementia, initiating monotherapy with one of the four antipsychotics of interest were eligible. Patients must also have been registered at the primary care practice for at least 6 months before their first prescription and eligible for HES/ONS data linkage (i.e., their practice participated in the linkage scheme, patient had valid identifiers and had not opted-out of data use for research).

Eligibility criteria were applied as at the first prescription (index) date, which is when follow-up commenced. Prescription for another antipsychotic at index (or a long-acting injectable in the prior 90 days) and first prescriptions labelled *pro re nata* were not considered consistent with initiating oral monotherapy for SMI.

Patients could only be eligible once (i.e., at first prescription). We allowed for SMI diagnosis to be recorded up to 30 days after index as these diagnoses are typically made by specialists in secondary care and subsequently communicated to primary care with potential delay. The clinician-verified SMI code list is publicly available[21]. The recruitment period was 1 January 2005 to 31 December 2014, allowing for 5 years follow-up prior to the COVID-19 pandemic (final follow-up, 31 December 2019).

### Exposures
In intention-to-treat analyses, the intervention was initiating aripiprazole as oral antipsychotic monotherapy, with olanzapine, quetiapine, or risperidone as comparators. In per-protocol analyses, the interventions were strategies of initiating and continuing (in the absence of

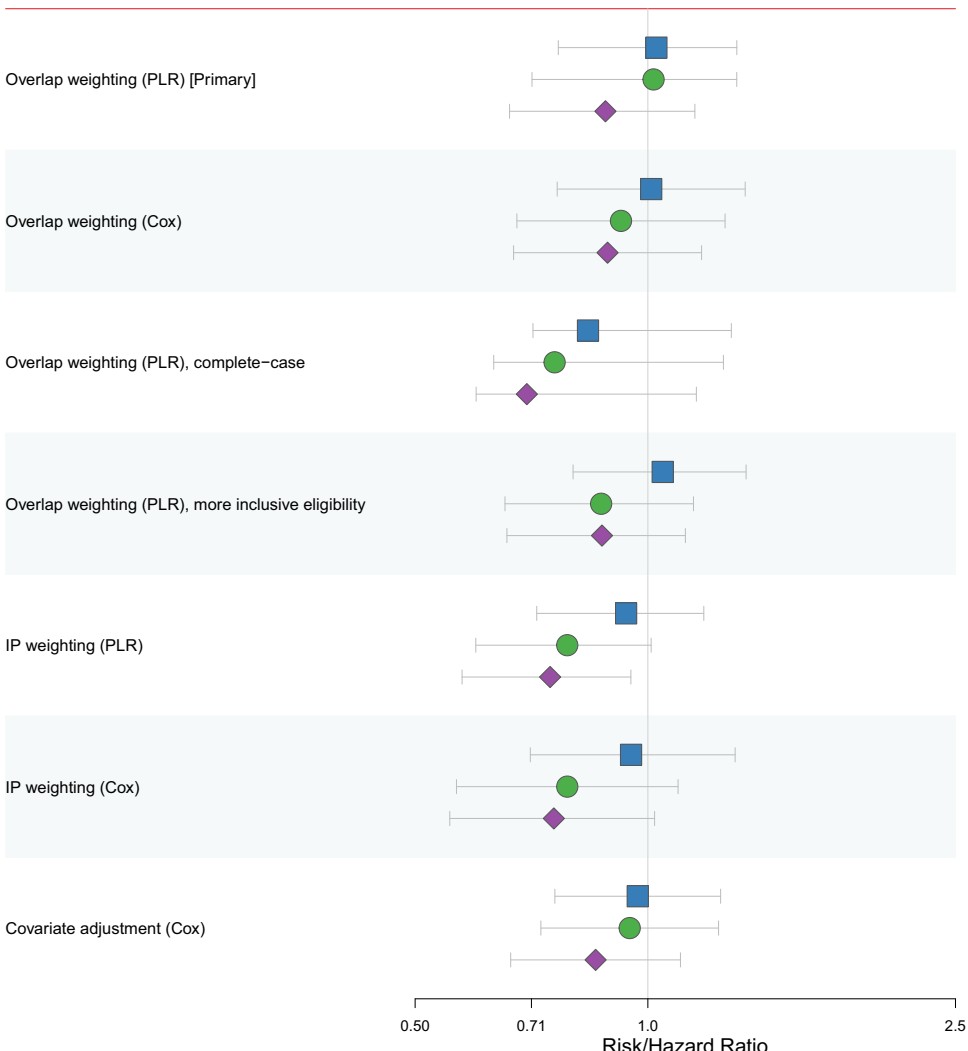

**Fig. 4 | Forest plot of intention-to-treat sensitivity analyses for the five-year risk of MACE.** MACE major adverse cardiovascular event, PLR pooled logistic regression, IP inverse probability. Estimates are risk or hazard ratios (as appropriate), with 95% confidence intervals (CIs). Risk ratios from the primary analysis (overlap weighting [PLR]) is shown for comparison. CIs for PLR results were derived from the distribution of 2500 estimates obtained via bootstrapping (i.e., 100 samples per imputed dataset). Bootstrap samples for the IP weighted PLR were increased to 500 due to instability at 100. Source data are provided as a Source Data file.

a recorded adverse reaction) monotherapy. In all analyses, initiation was defined according to the first prescription date, and dosing was as prescribed in clinical practice, with adjustment for starting daily dose.

Antipsychotics, identified using product codes[8], could be initiated by general practitioners or specialists (e.g., psychiatrist), but prescriptions must have been issued through primary care— standard practice for ongoing community management of SMI in the UK[22].

## Outcomes

The primary outcome was the 5-year risk of MACE—a composite of hospitalisation for non-fatal acute myocardial infarction or stroke and cardiovascular death, commonly used in cardiovascular research[15]. Secondary outcomes were the 6-month risk of MACE and 5-year risk of individual MACE components.

Using HES data, we defined acute myocardial infarction and stroke as hospitalisation events where the primary diagnosis during an episode of care was coded with International Classification of Diseases 10th Revision (ICD-10) codes (I21* and I60*-I64*, respectively), capturing diagnoses on or during hospital admission. Assessed against registry ascertainment, acute myocardial infarction recording in HES demonstrated a positive predictive value of 92%[23]. Against a resource-intensive biomedical evaluation in a large cohort, stroke recording in HES demonstrated a positive predictive value of 79%[24]. We defined cardiovascular death as ONS-registered deaths with an underlying cause coded to the 'Diseases of the circulatory system' ICD-10 chapter. Deaths within 28 days of myocardial infarction or stroke were considered fatal and coded as cardiovascular death.

## Covariates

Based on prior research and clinical expertise, and using a directed acyclic graph (Supplementary Fig. 4), we pre-specified 33 baseline covariates as candidate confounders. These included demographics, psychiatric history, comorbidities, and prescribed concomitant medications (see Supplemental Methods and Supplementary Table 7 for details). Sex and ethnicity were analysed as recorded in patient medical records. Where a patient had multiple ethnicity categories recorded, the most frequently recorded was used, or the most recent, if frequencies were equal.

## Statistical analysis

**Power calculation.** We considered a 10% relative difference in the primary outcome clinically important, given the severity of MACE. Assuming a 7.1% 5-year control event rate based on historical data, our a priori power calculation indicated that 2,048 patients across each pairwise comparison would provide 90% power to detect this difference ($p < 0.05$) (see Supplementary Methods for details).

**Missing data.** We assumed no missing outcomes given inclusion of patients eligible for linkage and the national coverage of linked datasets. Assuming missing at random, and using the *mice* package, we handled missing covariates (Supplementary Table 8) by generating 25 datasets using multiple imputation by chained equations (predictive mean matching for continuous and multinomial logistic regression for categorical variables).

**Primary analyses.** Our primary analyses compared aripiprazole with each comparator antipsychotic and followed the observational analogue of intention-to-treat—with patients analysed according to their original antipsychotic, irrespective of adherence.

To minimise confounding by measured covariates, we used propensity score generalised overlap weighting[14] to weight analyses to a target population with the greatest overlap in baseline covariates across all groups. This method is efficient[14] and includes data from all participants, but down-weights those with more extreme propensities (e.g., almost always/never treated). We computed normalised, generalised overlap weights as the inverse probability of treatment with the original antipsychotic, divided by the sum of the inverse probabilities of treatment with each antipsychotic[14]. We derived probabilities using multinomial logistic regression using the *nnet* package. To determine whether weighting achieved covariate balance, we assessed standardised differences after weighting, against a threshold of >0.10 to indicate potentially meaningful imbalance. Non-linear functional forms and interactions were iteratively considered to improve balance as required (Supplementary Table 7). We planned to use doubly robust methods if imbalances remained.

For MACE, follow-up time was censored at the earliest of incident MACE, non-cardiovascular death, or 5 years post-index date. A similar approach was taken for MACE components, but with follow-up censored at component, rather than composite, incidence. We fitted a weighted pooled logistic regression model[25] to estimate the average monthly probability of each outcome under each strategy. We included a treatment by time product term to allow for time-varying hazards[26]. We predicted absolute risks and calculated differences and ratios. We pooled point estimates across the 25 imputed datasets using Rubin's rules. We derived bias-corrected and accelerated percentile-based 95% confidence intervals[27] from 12,500 estimates obtained by bootstrapping the procedure[28] for estimating weights and outcome models, using 500 samples (stratified by treatment) per imputed dataset.

**Supplementary analyses.** We used a pooled logistic model, without a treatment by time product term, to approximate HRs and additionally fitted a continuous-time weighted Cox model for comparison.

We explored primary outcome variation by baseline age, sex, ethnicity, SMI diagnosis, and olanzapine-equivalent[29] starting daily dose, and considered Wald test *p*-values < 0.05 from comparing confounder-weighted Cox models with and without the relevant interaction terms as indicative of effect modification.

In per-protocol analysis, MACE follow-up time was additionally censored at the earliest of: antipsychotic switch, addition of another antipsychotic, discontinuation (i.e., 6 months[30] following last prescription), or, due to requiring prescription records to ascertain per-protocol status, de-registration from primary care or the practice last data collection date. Follow-up time was not censored if, within 30 days of switching, adding, or discontinuing the original antipsychotic, the patient had a recorded code indicating an adverse reaction to the original antipsychotic (e.g. "adverse reaction to quetiapine"), as changing treatment in this scenario was strategy consistent. We did not evaluate secondary outcomes in per-protocol analyses due to low absolute event numbers.

**Sensitivity analyses.** To explore the robustness of the primary outcome results to alternative approaches for handling baseline confounding, we conducted inverse probability of treatment weighting (with stabilised weights trimmed at the 99th percentile) applied to a pooled logistic and a Cox model. We fitted a Cox model with covariate adjustment. We conducted complete case analysis to examine the impact of missing covariates.

To explore selection bias due to censoring in the per-protocol analysis, we conducted inverse probability of censoring weighting. We derived weights (stabilised and trimmed at the 99.5th percentile) from the time-dependent probability of censoring. We combined censor weights with the generalised overlap weights and applied them to the pooled logistic model. In one model, censor weights were estimated based on pre-specified baseline covariates only; a second additionally included a monthly time-varying cumulative count of comorbidities during follow-up (Supplementary Table 7).

Finally, we analysed the primary outcome using more inclusive eligibility, allowing for SMI diagnoses to be recorded at any point before or during the study period. In primary analyses, eligibility criteria mandated that patients had an SMI diagnosis recorded at antipsychotic initiation (with an allowance for diagnoses up to 30 days after the index date), but we considered it plausible that most patients ultimately diagnosed with SMI during the study period were prescribed study antipsychotics for SMI symptoms.

Statistical tests were two-sided ($p < 0.05$, no adjustment for multiple comparisons)[31]. Analyses were conducted using R (version 4.4.0) in RStudio[32]. Unless otherwise indicated, to minimise computation time for subgroup and sensitivity analyses —where required, we derived percentile-based confidence intervals from 2500 estimates obtained via bootstrapping (i.e., 100 samples per imputed dataset).

**Ethical approval.** Ethical approval was obtained from the Independent Scientific Advisory Committee of CPRD (protocol no. 21_000729).

## Reporting summary

Further information on research design is available in the Nature Portfolio Reporting Summary linked to this article.

## Data availability

This study's data were accessed via Clinical Practice Research Datalink (CPRD) under approved protocol number 21_000729. The raw data are protected and are not available due to data privacy laws. However, data can be accessed from Clinical Practice Research Datalink (CPRD) following approval and licensing (see https://cprd.com/ for details). Source data are provided with this paper.

## Materials availability

Analytical code is available in the online repository: https://github.com/Alvin-RB/antipsychotics_mace_cprd. Correspondence and material requests should be addressed to alvin.richards-belle.21@ucl.ac.uk.

## Code availability

Analytical code is available in the online repository: https://github.com/Alvin-RB/antipsychotics_mace_cprd[32].

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

## Acknowledgements

This study is based in part on data from the Clinical Practice Research Datalink obtained under licence from the UK Medicines and Healthcare products Regulatory Agency. The data is provided by patients and collected by the NHS as part of their care and support. The interpretation and conclusions contained in this study are those of the author/s alone. Hospital Episode Statistics (HES) and Office for National Statistics (ONS), Copyright © (2024), re-used with the permission of The Health & Social Care Information Centre. All rights reserved. We thank Al Richards and Marianne West for advising on this research, all of the patients and primary care practices contributing to the Clinical Practice Research Datalink databases, and Clinical Practice Research Datalink for providing access to the data. ARB is funded by the Wellcome Trust through a PhD Fellowship in Mental Health Science (218497/Z/19/Z). This research was funded in whole or in part by the Wellcome Trust. For the purpose of Open Access, the author has applied a CC BY public copyright licence to any Author Accepted Manuscript (AAM) version arising from this submission. NL is supported by a Health Data Research UK personal

fellowship. This work is affiliated to Health Data Research UK (Big Data for Complex Disease - HDR-23012), which is funded by the Medical Research Council (UKRI), the National Institute for Health Research, the British Heart Foundation, Cancer Research UK, the Economic and Social Research Council (UKRI), the Engineering and Physical Sciences Research Council (UKRI), Health and Care Research Wales, Chief Scientist Office of the Scottish Government Health and Social Care Directorates, and Health and Social Care Research and Development Division (Public Health Agency, Northern Ireland). KKCM reports grants from the CW Maplethorpe Fellowship, the European Union Horizon 2020 (965381), the UK National Institute of Health Research (NIHR206166, NIHR201481), the Hong Kong Research Grant Council (17126722, 17114921, C7154-20G, 17112020, C7009-19GF), the Hong Kong Innovation and Technology Commission, and reports personal fees from IQVIA, unrelated to the current work. NMD is supported via Norwegian Research Council (295989). EB acknowledges the support of: Medical Research Council (G1100583, MR/W020238/1, MR/Z504816/1), National Institute of Health Research (NIHR200756), Mental Health Research UK - John Grace QC Scholarship 2018 & Economic Social Research Council's Co-funded doctoral award (ES/P000592/1), The British Medical Association's Margaret Temple Fellowship 2016, Medical Research Council New Investigator and Centenary Awards (G0901310, G1100583), and the University College London Hospitals NIHR Biomedical Research Centre. JFH is supported by UKRI grant MR/V023373/1, the University College London Hospitals NIHR Biomedical Research Centre and the NIHR North Thames Applied Research Collaboration. DPJO is supported by the University College London Hospitals NIHR Biomedical Research Centre and the NIHR North Thames Applied Research Collaboration. The views expressed in this article are those of the authors and not necessarily those of the NHS, the NIHR, or the Department of Health and Social Care. The funders had no role in study design, data collection and analysis, decision to publish, or preparation of the manuscript.

## Author contributions

A.R.B., E.B., J.F.H., and D.P.J.O. formulated the research questions and designed the study. A.R.B. analysed the data. A.R.B. wrote the first draft of the manuscript, and N.L., S.H., K.K.C.M., N.M.D., E.B., J.F.H. and D.P.J.O. critically reviewed the manuscript for important intellectual content. A.R.B., N.L., S.H., J.F.H. and D.P.J.O. had full access to the data. All authors approved the final version to be published and agree to be accountable for all aspects of the work in ensuring that questions related to the accuracy or integrity of any part of the work are appropriately investigated and resolved.

## Competing interests

JFH has received consultancy fees from Wellcome Trust, juli Inc and Swiss Re unrelated to the results of the current study. All other authors declare no competing interests.
