## [Transparent Peer Review file · Nature Communications]

Risk of major adverse cardiovascular events with aripiprazole versus olanzapine, quetiapine, and risperidone in severe mental illness: a target trial emulation

Corresponding Author: Mr Alvin Richards-Belle

Version 0:

Reviewer comments:

Reviewer #1

(Remarks to the Author)

Major comments

- 1) I think the main analysis for per-protocol should include adjustment for the introduced selection bias due to censoring because it is highly unlikely that such censoring is completely at random. The unadjusted results can then be reported in the appendix in case a reader is interested in exploring the magnitude and direction of such bias
- 2) I strongly suggest conducting a positive control outcome analysis using weight change as the outcome. It looks like the authors are suggesting that there is strong evidence on this outcome from randomized trials and this could help strengthen the case for a target trial emulation.
- 3) The authors have chosen the PS weighting approach to improve efficiency but the comparison of the main and sensitivity results shows that the IPW estimates are more efficient (and therefore imply a significant reduction in MACE risk for the comparison of Aripiprazole vs risperidone even under ITT). So, I think this at least needs to be mentioned in the results and discussed as well.
- 4) Line 363: the grace period of 6 months used for the per-protocol analysis seems too long, unless a typical prescription covers more than 3 months of treatment. I would suggest using a shorter grace period, say 2-3 months, in a sensitivity analysis.

Minor comments

- 5) Introduction: line 56: please clarify if reference #7 is from randomized trials or observational studies or both.
- 6) Methods, line 279: please clarify if the data on initiation is from prescription or if it also includes whether the patients filled the prescription.
- 7) Line 339: I think the Supplementary Table referred to here should be number 5 not 2.
- 8) Line 383: please clarify the lookback period for the number of comorbidities.
- 9) Line 383: why not also adjust for time-varying number of visits and hospitalizations in the past 3-6 months as a time-varying covariates.
- 10) Line 385-6: I don't understand the logic of removing the SMI Dx as an eligibility criterion. Why would a patient receive antipsychotics if they are not diagnosed with a severe mental illness. Please clarify.

(Remarks on code availability)

Reviewer #2

(Remarks to the Author)

The cohort study (20,404 patients) found that patients initiating aripiprazole in real-world conditions had similar five-year risks of MACE as those initiating olanzapine, quetiapine, and risperidone. Our per-protocol analysis showed that patients continuing aripiprazole had a lower risk than those continuing risperidone. This study is well-conducted, and the paper is well-written. It is easy for the reader to interpret the results of the paper.

The risk of MACE may be associated with severity of mental illness but not kind of antipsychotics. OLA and RIS are more

effective in severely ill patients with schizophrenia than ARI (PMID: 37526675). These associations should be examined by a future study including polygenic risk score.

(Remarks on code availability)

Reviewer #3

(Remarks to the Author)

(Remarks on code availability)

The cohort study (20,404 patients) found that patients initiating aripiprazole in real-world conditions had similar five-year risks of MACE as those initiating olanzapine, quetiapine, and risperidone. Our per-protocol analysis showed that patients continuing aripiprazole had a lower risk than those continuing risperidone. This study is well-conducted, and the paper is well-written. It is easy for the reader to interpret the results of the paper.

The risk of MACE may be associated with severity of mental illness but not kind of antipsychotics. OLA and RIS are more effective in severely ill patients with schizophrenia than ARI (PMID: 37526675). These associations should be examined by a future study including polygenic risk score.

Reviewer #4

(Remarks to the Author)

This is a nicely designed and executed target trial emulation regarding an important question. The paper is clearly written, the results are believable, the limitations of the methods clearly acknowledged.

The authors' interpretation of the results ("Patients initiating aripiprazole in real-world conditions had similar five-year risks of MACE as those initiating olanzapine, quetiapine, and risperidone."), however, seem to go beyond the study findings:

Abstract: "Targeting other potential modifiable causes of cardio-cerebrovascular risk might have greater impact on reducing risk."

Conclusion: "Antipsychotic choice might therefore play a relatively minor role in the elevated cardio-cerebrovascular risk observed in SMI."

The internal validity of the study seems high--for patients at clinical equipoise, antipsychotic choice doesn't make much difference in 5-year cardiovascular outcomes. And it is true that smoking, obesity, hypertension, and glucose intolerance might be better targets than antipsychotic choice.

However, for some of the large number of people included in the study (20,040 in the CONSORT chart) but not in the overlap weighted population (Table 1) who are presumably not at clinical equipoise, antipsychotic choice may well make a difference. We cannot tell from this study. It is possible that obese people who are channeled to aripiprazole may benefit from that choice. This limitation deserves mention.

Instead of the implications above, I propose something like:

Abstract: "For patients at clinical equipoise, antipsychotic selection does not appear to have significant effects on risk of the most severe, long-term cardiovascular events."

Conclusion: "For many patients with SMI, antipsychotic choice may not be an important way to reduce cardio-cerebrovascular risk. This study does not provide information on the effects of antipsychotic choice among those most at risk."

(Remarks on code availability)

Version 1:

Reviewer comments:

Reviewer #1

(Remarks to the Author)

None

(Remarks on code availability)

None

Re: NCOMMS-25-22796 “Risk of major adverse cardiovascular events with aripiprazole versus olanzapine, quetiapine, and risperidone in severe mental illness: a target trial emulation”

REVIEWER COMMENTS

Reviewer #1

Major comments

1) I think the main analysis for per-protocol should include adjustment for the introduced selection bias due to censoring because it is highly unlikely that such censoring is completely at random. The unadjusted results can then be reported in the appendix in case a reader is interested in exploring the magnitude and direction of such bias.

Response: We agree that the per-protocol analysis incorporating adjustment for potential selection bias is of particular importance given the risk of differential informative censoring. However, we wish to adhere to our pre-registered protocol in which we did not pre-specify this particular adjusted analysis as the primary analysis method for the per-protocol outcomes. Noting that these adjusted analyses were previously only mentioned briefly in the main text (with readers signposted to the supplementary materials for details) and to address this comment, we now describe

the analysis adjusting for potential selection bias due to censoring in more detail in the main Results (tracked version, page 6, line 167): “*Consistent with primary per-protocol analyses, all sensitivity analyses, including with different censoring weight approaches, found a lower five-year MACE risk with continuing aripiprazole versus risperidone. Using inverse probability of censoring weighting (with weights estimated based on baseline covariates and a monthly time-varying count of cumulative comorbidities during follow-up), the risk difference for patients continuing aripiprazole versus risperidone was -26.0 events per 1,000 (95% CI, -35.0 to -6.5), while risks remained similar when compared to continued olanzapine (-9.6 events, 95% CI, -20.2 to 4.1) and quetiapine (-2.1 events, 95% CI, -17.3 to 9.3) use.*”

While the results from the per-protocol analysis incorporating this adjustment were ultimately similar to the primary results, we believe that this re-organisation now makes the potential impact of selection bias clearer and allows readers to compare the estimates within the main results section. We hope this addresses the reviewer’s point.

2) I strongly suggest conducting a positive control outcome analysis using weight change as the outcome. It looks like the authors are suggesting that there is strong evidence on this outcome from randomized trials and this could help strengthen the case for a target trial emulation.

Response: We thank the reviewer for this suggestion. In a previous target trial emulation, we studied the effect of the same four antipsychotics on medium-term cardiometabolic and psychiatric effectiveness outcomes. We replicated the well-established finding from randomised trials that olanzapine leads to significantly greater weight gain compared to aripiprazole.[1] This prior study demonstrated that this methodology can successfully be implemented to produce clinically important and credible results, specifically in the setting of antipsychotics for SMI. Although not directly a positive control outcome for the present MACE analysis, we believe that our prior work strengthens the case for the present target trial emulation (as the reviewer suggests) where we apply a similar methodology to the study of an important, but understudied, longer-term outcome.

We had previously cited this prior work, but given its relevance, in response to the reviewer’s comment we have now expanded its discussion in the Introduction (page 3, line 91): “*We built on earlier work demonstrating the feasibility of emulating RCTs*

of antipsychotics in SMI using linked UK data sources and which generated clinically important and credible real-world evidence, including replication of the established weight gain observed with olanzapine versus aripiprazole.[1]"

We also now refer to this work in the Discussion (page 8, line 246): "*Second, building on prior work demonstrating feasibility,[1] we leveraged the target trial emulation framework...*"

3) The authors have chosen the PS weighting approach to improve efficiency but the comparison of the main and sensitivity results shows that the IPW estimates are more efficient (and therefore imply a significant reduction in MACE risk for the comparison of Aripiprazole vs risperidone even under ITT). So, I think this at least needs to be mentioned in the results and discussed as well.

Response: We thank the reviewer for highlighting this. In our primary analyses, we used overlap weighting to estimate the average treatment effect in the target population of patients with the greatest overlap in baseline covariates (i.e., a population at clinical equipoise). As noted by the reviewer, we reported inverse probability of treatment weighting (IPTW) estimates as a sensitivity analysis.

While relevant and of interest, the estimand for IPTW is the average treatment effect in the overall population, and although the IPTW estimates were more efficient, they are not directly comparable to those from the primary analysis for this reason. Additionally, the different weighting schemes lead to different effective sample sizes impacting precision. Further, IPTW typically requires ad-hoc trimming methods to handle extreme weights, an issue avoided with overlap weighting.

While we identified some significant reductions in MACE risk in some IPTW pooled logistic regression results, evidence for these effects was less strong when estimated using a weighted Cox model. These inconsistent results suggest sensitivity to model specification, and, for this reason, we do not place significant weight on these findings. To address this comment, we are now more precise (with regards to the estimand) when describing the IPTW results, and we highlight in the main text the discrepancy between the IPTW pooled logistic and Cox model results:

Intention-to-treat results (page 5, line 162) – "*The exception was the inverse probability of treatment weighting pooled logistic model results which indicated a*

lower five-year MACE risk with aripiprazole initiation versus risperidone in the overall population – however, evidence for this effect was less strong when estimated with a weighted Cox model.”

Per-protocol results (page 6, line 174) – *“The inverse probability of treatment weighting pooled logistic model results also indicated a lower risk with continuing aripiprazole versus quetiapine in the overall population, but with less strong evidence when estimated with a weighted Cox model.”*

4) Line 363: the grace period of 6 months used for the per-protocol analysis seems too long, unless a typical prescription covers more than 3 months of treatment. I would suggest using a shorter grace period, say 2-3 months, in a sensitivity analysis.

Response: In our per-protocol analysis, we clarify in the Methods that we censored patient follow-up time at the earliest of the following:

- incident MACE,
- non-cardiovascular death,
- five years post-index date,
- antipsychotic switch,
- addition of another antipsychotic,
- discontinuation (i.e., six months^[2] following last prescription), or,
- due to requiring prescription records to ascertain per-protocol status, de-registration from primary care or the practice last data collection date.

For the discontinuation criteria, we selected a 6-month grace period based on Douglas & Smeeth (2008), who reported data in a UK primary care sample to support the statement that *“the risk of stroke might not fall to baseline until six months after a patient stops taking an antipsychotic drug.”*^[2] We cite this article in the main text.

While shorter grace periods (e.g., 2-3 months) could be explored, there is less empirical evidence supporting these durations for cardiovascular events and they may be comparatively less sensitive for capturing events if risk remains elevated for several months following antipsychotic discontinuation. Evaluating multiple discontinuation grace periods within per-protocol analyses was beyond the scope of the present study; however, delineating temporality of risk post-discontinuation is an

important area for future research, which we now mention in the Discussion to address the reviewer's point (page 7, line 219): *"Future research should also examine the effects of dose variation and treatment duration, as well as delineate temporality of risk post-discontinuation, particularly since olanzapine and quetiapine are typically thought to confer greater cardiovascular risk than risperidone."*

Minor comments

5) Introduction: line 56: please clarify if reference #7 is from randomized trials or observational studies or both.

Response: Reference #7 is a systematic review and meta-analysis of observational cohort studies. We now clarify this in the Introduction (page 3, line 71): *"A comprehensive meta-analysis of observational cohort studies reported that use of second-generation antipsychotics (versus non-use) is protective overall for cardio-cerebrovascular mortality."*

6) Methods, line 279: please clarify if the data on initiation is from prescription or if it also includes whether the patients filled the prescription.

Response: We thank the reviewer for highlighting this omission – initiation was defined based on prescription date. We now clarify this in the Methods (page 11, line 336): *"In all analyses, initiation was defined according to the first prescription date."*

Finally, we have added the lack of dispensing data as a limitation in the Discussion (page 8, line 257): *"Second, we did not have data on medication dispensation or adherence and so relied on prescription records to define treatment strategies."*

7) Line 339: I think the Supplementary Table referred to here should be number 5 not 2.

Response: On page 13, line 395 (Methods), we discuss consideration of functional forms improve covariate balance – of which the forms are detailed in Supplementary Table 2. Supplementary Table 5 is referred to in the Results (page 4, line 123). Finally, we have reviewed all references to tables and figures to ensure that they are correct throughout.

8) Line 383: please clarify the lookback period for the number of comorbidities.

Response: For baseline comorbidities, we looked back over the patient's entire medical history (lookback periods are specified in the footnotes of Table 1). In line 383, we discussed using a time-varying cumulative count of comorbidities within censoring weights. This variable was based on comorbidities recorded in patient's medical records in each month of follow-up. To address this, we have clarified this detail in the text (page 14, line 438): "...a second additionally included a monthly time-varying cumulative count of comorbidities during follow-up."

9) Line 383: why not also adjust for time-varying number of visits and hospitalizations in the past 3-6 months as time-varying covariates.

Response: We thank the reviewer for this thoughtful suggestion. In designing our inverse probability of censoring weighting approach for the per-protocol sensitivity analysis, we aimed to construct a parsimonious model that captured potential health deterioration during follow-up which could have impacted censoring. We balanced this aim against the need to avoid overfitting, particularly given the large number of variables included in the model and our overall modest sample size (although large for this clinical context).

We considered including time-varying counts of visits and hospitalisations (as the reviewer suggests), but ultimately favoured cumulative comorbidity counts for several reasons. First, visit and hospitalisation counts are subject to additional biases relating to health-seeking behaviours and may be less robust markers of censoring risk. Second, our comorbidity counts (representing nine important and pre-specified comorbidities) were derived from the primary care and hospitalisation records, potentially capturing the underlying health deterioration identified in these encounters. Including both would therefore have created substantial collinearity. Finally, we note that baseline numbers of GP visits and hospitalizations were pre-specified covariates, and so health-seeking propensity was already captured in our models, albeit without heavy weighting given the potential biases outlined above.

Regarding the timeframe, we estimated weights in monthly intervals during follow-up. Instead of considering recent diagnoses only (e.g., over the past 3-6 months as the reviewer suggested), we carried forward the diagnoses until the end of follow-up given that the comorbidities were primarily chronic conditions and their accumulation over time is also relevant to censoring risk.

10) Line 385-6: I don't understand the logic of removing the SMI Dx as an eligibility criterion. Why would a patient receive antipsychotics if they are not diagnosed with a severe mental illness. Please clarify.

Response: In our primary analysis, as an eligibility criterion we required an SMI diagnosis to be recorded in the patient's primary care record at any time before or up to 30 days after the first antipsychotic prescription.

In the sensitivity analysis, we relaxed this window to allow the diagnosis to be recorded at any time during the study period. This decision was based on clinical experience that some patients are prescribed antipsychotics before the formal diagnosis is coded and recognising delays (beyond 30 days) in communicating information from secondary care to primary care records (our data source).

Importantly, even with the broader window, we still required an SMI diagnosis to be recorded within the study period to reduce the risk of including patients prescribed antipsychotics for other indications (e.g., anxiety, depression, personality disorders, sleep disorders), which can be relatively common in primary care.[3]

We have added further detail on the rationale for this analysis in the Methods (page 14, line 442): *"Finally, we analysed the primary outcome using more inclusive eligibility, allowing for SMI diagnoses to be recorded at any point before or during the study period. In primary analyses, eligibility criteria mandated that patients had an SMI diagnosis recorded at antipsychotic initiation (with an allowance for diagnoses up to 30 days after the index date), but we considered it plausible that most patients ultimately diagnosed with SMI during the study period were prescribed study antipsychotics for SMI symptoms."*

Reviewer #2 and #3

The cohort study (20,404 patients) found that patients initiating aripiprazole in real-world conditions had similar five-year risks of MACE as those initiating olanzapine, quetiapine, and risperidone. Our per-protocol analysis showed that patients continuing aripiprazole had a lower risk than those continuing risperidone. This study is well-conducted, and the paper is well-written. It is easy for the reader to interpret the results of the paper.

Response: We thank the reviewers for their positive comments on our study.

The risk of MACE may be associated with severity of mental illness but not kind of antipsychotics. OLA and RIS are more effective in severely ill patients with schizophrenia than ARI (PMID: 37526675). These associations should be examined by a future study including polygenic risk score.

Response: Although beyond the scope of the present study, we agree that exploring the association between the severity of mental illness and the risk of MACE is an important future research area which we now mention in the Discussion (page 7, line 203): "*Exploring the association between the severity of mental illness and cardio-cerebrovascular risk is also an important area for future research.*"

Response: We thank the reviewers for their time on the co-review.

Reviewer #4

This is a nicely designed and executed target trial emulation regarding an important question. The paper is clearly written, the results are believable, the limitations of the methods clearly acknowledged.

Response: We thank the reviewer for their positive comments on our study.

The authors' interpretation of the results ("Patients initiating aripiprazole in real-world conditions had similar five-year risks of MACE as those initiating olanzapine, quetiapine, and risperidone."), however, seem to be beyond the study findings:

Abstract: "Targeting other potential modifiable causes of cardio-cerebrovascular risk might have greater impact on reducing risk."

Conclusion: "Antipsychotic choice might therefore play a relatively minor role in the elevated cardio-cerebrovascular risk observed in SMI."

The internal validity of the study seems high--for patients at clinical equipoise, antipsychotic choice doesn't make much difference in 5-year cardiovascular outcomes. And it is true that

smoking, obesity, hypertension, and glucose intolerance might be better targets than antipsychotic choice.

However, for some of the large number of people included in the study (20,040 in the CONSORT chart) but not in the overlap weighted population (Table 1) who are presumably not at clinical equipoise, antipsychotic choice may well make a difference. We cannot tell from this study. It is possible that obese people who are channeled to aripiprazole may benefit from that choice. This limitation deserves mention.

Instead of the implications above, I propose something like:

Abstract: "For patients at clinical equipoise, antipsychotic selection does not appear to have significant effects on risk of the most severe, long-term cardiovascular events."

Conclusion: "For many patients with SMI, antipsychotic choice may not be an important way to reduce cardio-cerebrovascular risk. This study does not provide information on the effects of antipsychotic choice among those most at risk."

Response: We thank the reviewer for their comments and close attention to these important details.

We agree with the reviewer that it is possible that antipsychotic choice does matter for MACE risk among patients not at clinical equipoise. Although overlap weighting still leverages data from all patients, weights are proportionally down weighted as patient's covariate distribution moves further away from the area of overlap. It is therefore crucial to interpret our results with reference to the target population. As our target population in this study was patients at clinical equipoise, we are not able to directly provide evidence for or comment on outcomes for patients not at clinical equipoise. This limitation is however not unique to our study, as it commonly arises in randomised trials – whereby patients not at clinical equipoise are not recruited and randomised. We now address this limitation in the Discussion (page 9, line 262): *"Fourth, our focus on patients at clinical equipoise precludes inference on patients not at clinical equipoise for whom risks might differ."*

We also thank the reviewer for their suggested wordings for the Abstract and Conclusion which we have now used and further refined as follows:

Abstract Conclusion – “*For patients at clinical equipoise, antipsychotic selection does not appear to have significant effects on risk of the most severe, long-term cardiovascular events. However, further research is needed to replicate our finding of increased MACE risk with the continued use of risperidone compared to aripiprazole.*”

Paper Conclusion – “*For patients at clinical equipoise for selection of second-generation antipsychotic, those initiating aripiprazole in real-world conditions had similar five-year risks of MACE as those initiating olanzapine, quetiapine, and risperidone. Therefore, for many patients, antipsychotic selection does not appear to have significant effects on risk of the most severe, long-term cardiovascular events. We need further research to replicate our finding of reduced risk in patients continuing aripiprazole versus risperidone and to elucidate underlying mechanisms.*”

References

- [1] A. Richards-Belle *et al.*, ‘Comparative cardiometabolic safety and effectiveness of aripiprazole in people with severe mental illness: A target trial emulation’, *PLOS Medicine*, vol. 22, no. 1, p. e1004520, Jan. 2025, doi: 10.1371/journal.pmed.1004520.
- [2] I. J. Douglas and L. Smeeth, ‘Exposure to antipsychotics and risk of stroke: self controlled case series study’, *BMJ*, vol. 337, p. a1227, Aug. 2008, doi: 10.1136/bmj.a1227.
- [3] L. Marston, I. Nazareth, I. Petersen, K. Walters, and D. P. J. Osborn, ‘Prescribing of antipsychotics in UK primary care: a cohort study’, *BMJ Open*, vol. 4, no. 12, p. e006135, Dec. 2014, doi: 10.1136/bmjopen-2014-006135.